

# Treatment with sodium (*S*)-2-hydroxyglutarate prevents liver injury in an ischemia-reperfusion model in female Wistar rats

Eduardo Cienfuegos-Pecina[1,2], Diana P. Moreno-Peña[1], Liliana Torres-González[1], Diana Raquel Rodríguez-Rodríguez[1], Diana Garza-Villarreal[1], Oscar H. Mendoza-Hernández[1], Raul Alejandro Flores-Cantú[1], Brenda Alejandra Samaniego Sáenz[1], Gabriela Alarcon-Galvan[3], Linda E. Muñoz-Espinosa[1], Tannya R. Ibarra-Rivera[4], Alma L. Saucedo[4] and Paula Cordero-Pérez[1]

[1] Universidad Autónoma de Nuevo León. Liver Unit, Department of Internal Medicine, University Hospital "Dr. José E. González", Monterrey, Nuevo León, Mexico

[2] Universidad Autónoma de Nuevo León. Blood Bank, Department of Clinical Pathology, University Hospital "Dr. José E. González", Monterrey, Nuevo León, Mexico

[3] Universidad de Monterrey, Basic Science Department, School of Medicine, Monterrey, Nuevo León, Mexico

[4] Universidad Autónoma de Nuevo León. Department of Analytical Chemistry, School of Medicine, Monterrey, Nuevo León, Mexico

Corresponding author
Paula Cordero-Pérez,
paula.corderoprz@uanl.edu.mx

## ABSTRACT

**Background**. Ischemia-reperfusion (IR) injury is one of the leading causes of early graft dysfunction in liver transplantation. Techniques such as ischemic preconditioning protect the graft through the activation of the hypoxia-inducible factors (HIF), which are downregulated by the EGLN family of prolyl-4-hydroxylases, a potential biological target for the development of strategies based on pharmacological preconditioning. For that reason, this study aims to evaluate the effect of the EGLN inhibitor sodium (*S*)-2-hydroxyglutarate [(*S*)-2HG] on liver IR injury in Wistar rats.

**Methods**. Twenty-eight female Wistar rats were divided into the following groups: sham (SH, $n = 7$), non-toxicity (HGTox, $n = 7$, 25 mg/kg of (*S*)-2HG, twice per day for two days), IR ($n = 7$, total liver ischemia: 20 minutes, reperfusion: 60 minutes), and (*S*)-2HG+IR (HGIR, $n = 7$, 25 mg/kg of (*S*)-2HG, twice per day for two days, total liver ischemia as the IR group). Serum ALT, AST, LDH, ALP, glucose, and total bilirubin were assessed. The concentrations of IL-1β, IL-6, TNF, malondialdehyde, superoxide dismutase, and glutathione peroxidase were measured in liver tissue, as well as the expression of *Hmox1*, *Vegfa*, and *Pdk1*, determined by RT-qPCR. Sections of liver tissue were evaluated histologically, assessing the severity of necrosis, sinusoidal congestion, and cytoplasmatic vacuolization.

**Results**. The administration of (*S*)-2HG did not cause any alteration in the assessed biochemical markers compared to SH. Preconditioning with (*S*)-2HG significantly ameliorated IR injury in the HGIR group, decreasing the serum activities of ALT, AST, and LDH, and the tissue concentrations of IL-1β and IL-6 compared to the IR group. IR injury decreased serum glucose compared to SH. There were no differences in the other biomarkers assessed. The treatment with (*S*)-2HG tended to decrease the severity of hepatocyte necrosis and sinusoidal congestion compared to the IR group. The administration of (*S*)-2HG did not affect the expression of *Hmox1* but decreased

the expression of both *Vegfa* and *Pdk1* compared to the SH group, suggesting that the HIF-1 pathway is not involved in its mechanism of hepatoprotection. In conclusion, (*S*)-2HG showed a hepatoprotective effect, decreasing the levels of liver injury and inflammation biomarkers, without evidence of the involvement of the HIF-1 pathway. No hepatotoxic effect was observed at the tested dose.

## INTRODUCTION

Orthotopic liver transplantation is the definitive therapy for either end-stage chronic liver disease or severe acute liver failure (*Bachir, Larson & Palmer, 2012*). In Mexico, by the first half of 2020, 317 patients were on a waiting list for liver transplantation and three patients were waiting for combined liver-kidney transplantation. During the same period, only 39 liver transplantations and one combined liver-kidney transplantation were performed in the country (*CENATRA, 2020*). Despite the improvements in the methodologies of HLA-typification and the immunosuppression strategies, there is still a significant incidence of early graft dysfunction, which is defined by a spectrum of clinical signs that end in acute graft rejection and the need for a retransplant (*Briceño & Ciria, 2010*). The leading cause of early graft dysfunction is the ischemia-reperfusion (IR) injury; a process that occurs when blood flow to the organ is impaired, causing long-time ischemia, and then it is suddenly restored, reperfusing the tissue with oxygenated blood. Paradoxically, the oxygen influx enhances the damage mechanisms triggered by hypoxia (*Salvadori, Rosso & Bertoni, 2015*).

IR injury is a complex process, involving a network of damage mechanisms triggered by hypoxia. When intracellular $O_2$ concentrations plunge, the electron transport chain stops, and the anaerobic glycolysis becomes the main source of ATP for the cell (*Dorweiler et al., 2007*). Meanwhile, as ATP is depleted, the concentration of ADP increases. ADP is metabolized to hypoxanthine, and its degradation is mediated by the xanthine dehydrogenase, an $NAD^+$/NADH oxidoreductase which activity is impaired during the ischemic period, increasing the hypoxanthine concentration by 10-fold after 2 h of ischemia (*Schoenberg et al., 1985*). When the tissue is suddenly reperfused with oxygenated blood, the oxidative environment favors the oxidation of several cysteine residues in the enzyme xanthine dehydrogenase, causing a conformational change that changes the activity of the enzyme to an oxidase (*Nishino et al., 2005*), producing, during the reperfusion, an oxidative burst that is usually lethal to the cells. The fact that reperfusion triggers most of the injury, instead of preventing it, is known as the paradox of IR (*Salvadori, Rosso & Bertoni, 2015*).

A diversity of strategies have been developed to ameliorate the IR injury, with a particular emphasis on techniques such as the ischemic preconditioning (IPC) (*Murry, Jennings & Reimer, 1986*) and the remote ischemic preconditioning (RIPC) (*Przyklenk et al., 1993*), which consist in short and intermittent periods of ischemia, either in the target organ or

in distal organs. It is known that the primary protective mechanism involved with these techniques is the hypoxia-inducible factor (HIF) pathway (*Albrecht et al., 2013*).

HIFs are transcription factors acting as the primary regulators of oxygen homeostasis. These are heterodimers, conformed by an α subunit, oxygen-dependent, and a β subunit constitutively expressed (*Semenza, 2014*). During normoxia, the α subunit is hydroxylated by the EGLN family of prolyl-4-hydroxylases, which are oxygen and α-ketoglutarate dependent dioxygenases, and act as the primary oxygen sensors in the cell. After hydroxylation, α subunits are ubiquitinated by the von Hippel-Lindau E3 ubiquitin ligase and degraded in the proteasome. On the other hand, during hypoxia, EGLN hydroxylases are inactive, and the α subunits get stabilized, translocated to the nucleus, and dimerized with β-subunits, favoring the transcription of their target genes (*Dengler, Galbraith & Espinosa, 2014*). For that reason, this metabolic pathway is a potential therapeutic target for the development of pharmacologic preconditioners.

The 2-hydroxyglutaric acid is an inhibitor of the EGLN hydroxylases. Its (*R*) enantiomer was the first oncometabolite described in the literature, found in tumors with high resistance to hypoxia, usually harboring a mutated isocitrate dehydrogenase 1 (*Dang et al., 2009*). On the other hand, its (*S*) enantiomer is naturally produced *in vivo* as a product of non-specific reactions of several enzymes, such as the malate dehydrogenase or the lactate dehydrogenase A (*Du & Hu, 2021*; *Intlekofer et al., 2017*). The (*S*)-2-hydroxyglutarate [(*S*)-2HG] has a higher inhibitory effect against EGLN-1 than against EGLN-2 or 3 (*Koivunen et al., 2012*), and a previous report of our research group showed that the oral administration of its disodium salt has a nephroprotective effect against IR injury in Wistar rats, suggesting the involvement of the HIF-1 pathway in its mechanism (*Cienfuegos-Pecina et al., 2020*). Given the pharmacologic potential of this compound, we aimed to evaluate whether the administration of (*S*)-2HG has a hepatoprotective effect against IR injury in Wistar rats.

## MATERIALS & METHODS

### Synthesis of (*S*)-2HG

(*S*)-2HG was synthesized using a two-step methodology from L-glutamic acid, by using the optimized synthetic route reported by our laboratory (*Cienfuegos-Pecina et al., 2020*). Briefly, L-glutamic acid was subjected to a diazotization reaction in the presence of $NaNO_2$ and $H_2SO_4$ for 24 h. The reaction was stopped with NaCl and extracted three times with ethyl acetate, obtaining the 5-oxotetrahydrofuran-2-carboxylic acid, which was purified by column chromatography, using silica as the stationary phase and ethyl acetate as the mobile phase. This compound was hydrolyzed with NaOH, pH 10 for two hours, obtaining (*S*)-2HG, which was precipitated with anhydrous methanol.

The identity of the synthesis product was confirmed by nuclear magnetic resonance (NMR) spectroscopy and polarimetry. NMR data were acquired in a Bruker AVANCE III HD 400 MHz spectrometer (Bruker Corp., Billerica, MA, USA). A 50 mg sample of (*S*)-2HG was dissolved in double-distilled water and analyzed using sodium 3-(trimethylsilyl)propionate-2,2,3,3-d$_4$ (TSP) in $D_2O$ (Sigma-Aldrich) as an internal standard. The $^1$H-NMR and $^{13}$C-NMR spectra were acquired using the *noesypr1d* pulse

sequence for water signal suppression and compared with those reported in the literature (*Bal & Gryff-Keller, 2002*; *Cienfuegos-Pecina et al., 2020*).

The specific optical rotation ($[\alpha]_D^{20°C}$) of the synthetic ($S$)-2HG was measured in a PerkinElmer 341 Polarimeter (PerkinElmer, Waltham, MA, USA). Data were compared with previous reports in the literature (*Cienfuegos-Pecina et al., 2020*; *Ritthausen, 1872*).

## Animals

Twenty-eight healthy female Wistar rats, weighing $238 \pm 18$ g, were used. Every single animal was considered an experimental unit. Rats were bred in-house, and they were kept under standard conditions of light and temperature ($24 \pm 3$ °C, 12 h light-dark cycles), with access to commercial rat pellets (Nutrimix de México, Mexico) and water *ad libitum*. The animals were not genetically engineered, and no previous procedures were performed on the animals before the experiments. All the procedures were performed according to the specifications of the Mexican Official Norm NOM-062-ZOO-1999. This project was submitted to the Ethics and Research Committee of the School of Medicine, Universidad Autónoma de Nuevo León, and approved with the register number HI19-00002.

## Experimental design

The sample size was decided based on the result of an *a priori* calculation using (1).

$$n = \frac{(Z\alpha + Z\beta)^2(\sigma_1^2 + \sigma_2^2)}{(\mu_1 - \mu_2)^2} \tag{1}$$

Equation (1). Estimation of the sample size for comparison of means. $Z\alpha$ = Z-value for $\alpha$; $Z\beta$ = Z-value for $\beta$; $\sigma_1$ = expected standard deviation of group 1; $\sigma_2$ = expected standard deviation of group 2; $\mu_1$ = expected mean of group 1; $\mu_2$ = expected mean of group 2.

As a reference, we considered a previously reported serum activity of ALT of $494 \pm 84$ U/L for rats subjected to the same IR-injury-induction protocol we used (*Jiménez Pérez et al., 2016*). We calculated the sample size expecting a 40% decreasing of the serum ALT activity after the treatment (with no change in the standard deviation) and considered a two-tail statistical significance ($\alpha$) of 0.01 and a statistical power ($1-\beta$) of 95%, obtaining a total of 7 rats per group.

Rats were paired with a random number sequence obtained in R v. 4.01. The randomized animals were assigned to the following groups:

- Sham group (SH), n = 7: Rats were treated with double distilled water, *p.o.*, twice per day, for two days. Then they were undergone to laparotomy with exposure of the liver hilum, without inducing IR injury. After 1 h and 20 min, rats were sacrificed by exsanguination, obtaining blood and tissue samples.
- Nontoxicity group (HGTox), n = 7: Rats were treated with double-distilled-water-dissolved ($S$)-2HG, at a dose of 25 mg/kg, in the same way as the SH group. After treatment, they underwent the same procedure as the SH group. A final n of 6 was considered for the data analysis since one of the rats suddenly died after the anesthesia was administered.

- IR group (IR), n = 7: Rats were treated in the same way as the SH group. Then they underwent laparotomy with an exposition of the liver hilum and induction of IR injury (20 min of ischemia, one hour of reperfusion).
- (S)-2HG + IR group (HGIR), n = 7: A dose of 25 mg/kg of (S)-2HG, was administered to the rats in the same way as the HGTox group. Then, they underwent the same procedure as the IR group.

To minimize potential confounders, all rats were housed in the same room during the experiments, under the same conditions of light and temperature. In each surgery, an equal number of rats from each group underwent the surgical procedure. During the analysis of the biological samples, all the involved were blinded to the identity of the samples, except those who then performed the statistical analysis (Eduardo Cienfuegos-Pecina and Paula Cordero-Pérez).

### Induction of liver injury

Ischemic liver injury was induced following the procedure reported by *Jiménez Pérez et al. (2016)*. Rats were anesthetized with 100 mg/kg of ketamine (Anesket, PiSA Agropecuaria, S.A. de CV, Mexico), and 10 mg/kg of xylazine (Sedaject, Vedilab S.A. de CV, Mexico), *i. p*. After asepsis, an incision along the midline was performed, exposing the hepatic hilum, which was then occluded by using an atraumatic vascular clamp (Pringle maneuver) for 20 min. Following the ischemia, the clamp was withdrawn, and the rats were kept under general anesthesia for a reperfusion time of 1 h, after which rats were sacrificed by exsanguination by a puncture in the aorta. Criteria for early euthanasia was the failure of the anesthesia during the surgery, however, none of the animals needed the use of this protocol. Blood samples were centrifuged at 2,000 g for 12 min, then the serum was separated. The liver was resected, weighed, and samples of tissue were obtained and frozen at −80 °C until analysis.

### Biochemical markers, oxidative stress biomarkers, and proinflammatory cytokines

To assess the magnitude of the induced liver injury, the serum activities of alanine aminotransferase (ALT), aspartate aminotransferase (AST), lactate dehydrogenase (LDH), and alkaline phosphatase (ALP), and the serum concentrations of total bilirubin and glucose were measured. The biochemical determinations were performed in an ILab Aries analyzer (Instrumentation Laboratory, SpA, Milan, Italy), using kinetic and end-point UV-Visible spectrophotometric methodologies.

Malondialdehyde (MDA) is one of the main products of arachidonic acid peroxidation, and it is a commonly used biomarker to assess oxidative stress (*Ayala, Muñoz & Argüelles, 2014*), alongside the activities of superoxide dismutase (SOD) and glutathione peroxidase (GPx). Since one of the key mechanisms involved in IR injury is the oxidative burst produced by the sudden activation of xanthine oxidase, we compared the tissue levels of MDA, SOD, and GPx among the study groups. To measure the concentration of MDA in liver tissue, 200 mg of liver tissue were mechanically homogenized in 1 mL of RIPA buffer in an ice bath, and then centrifuged at 1,600 g for 10 min at 4 °C. MDA concentration
was measured spectrophotometrically in the supernatant, using the thiobarbituric acid colorimetric method with a thiobarbituric acid-reactive substances (TBARS) assay kit (Cayman Chemical Company, Ann Arbor, MI, USA). The product of this reaction was measured spectrophotometrically at a wavelength of 535 nm.

The total tissue activity of SOD was quantified by measuring the inhibition of the reduction of a tetrazolium salt to formazan by reactive oxygen species using the SOD Assay Kit (Cayman Chemical Company). Briefly, 200 mg of liver tissue were mechanically homogenized in one mL of 20 mM HEPES buffer, pH 7.2, containing 1 mM EDTA 210 mM mannitol, and 70 mM sucrose. Homogenization was done in an ice bath. Samples were then centrifuged for 15 min at 10,000 g, at 4 °C and the assay was performed using the supernatant, diluted in the buffer. 96-well microplates were read spectrophotometrically at a wavelength of 450 nm and total SOD activity was determined.

Tissue GPx activity was measured by quantifying the oxidation of NADPH to $NADP^+$ using a GPx Assay Kit (Cayman Chemical Company). To perform the assay, 200 mg of liver tissue were mechanically homogenized in one mL of 50 mM Tris-HCl buffer, pH 7.5, containing 5 mM EDTA and 1 mM DTT. Homogenization was performed in an ice bath. Samples were centrifuged at 10.000 g for 15 min at 4 °C, and the supernatants were separated. The supernatants were diluted in the sample buffer provided by the manufacturer before the GPx determination. GPx activity was quantified by a UV kinetic method, measuring the change in the absorbance at a wavelength of 340 nm.

To assess the inflammatory response after the induction of IR injury, we measured the tissue concentrations of the proinflammatory cytokines interleukin 1β (IL-1β), interleukin 6 (IL-6), and tumor necrosis factor (TNF) using a commercial sandwich enzyme-linked immunosorbent assay (ELISA) (R&D Systems, Minneapolis, MN, USA). The same samples from the GPx determination were used, undiluted. ELISA protocols were performed following the manufacturer's instructions. Cytokine concentrations were determined spectrophotometrically, at a wavelength of 450 nm.

All measurements in tissue were normalized to the protein concentration in the homogenates, measured using Bradford's method. Bradford's reagent was prepared as previously reported in the literature (*Kruger, 2009*). MDA concentration is reported as mmol/mg of protein, SOD activity is reported as IU/mg of protein, GPx activity is reported as nmol/min/mg of protein, and proinflammatory cytokines are reported as pg/mg of protein.

## Histological evaluation

Samples of liver tissue were fixed in phosphate-buffered 10% formalin solution pH 7.4 and then embedded in paraffin blocks, which were cut using a microtome at a thickness of 4 μm. The tissue sections were deparaffinized and processed using the standard histological technique. Hematoxylin-eosin (H&E) stained sections were blindly evaluated under the microscope, assessing the severity of necrosis, cytoplasmic vacuolization, and sinusoidal congestion by using the damage scale reported by *Susuki et al. (1993)*: 0 = no evident injury; 1 = single-cell necrosis, minimal congestion or vacuolization; 2 = necrosis <30%, mild

congestion or vacuolization; 3 = necrosis <60%, moderate congestion or vacuolization; and 4 = necrosis >60%, severe congestion or vacuolization.

## Quantitative RT-PCR

We used RT-qPCR to measure the expression of *Hmox1*, *Vegfa*, and *Pdk1*, the genes coding for heme-oxygenase 1, vascular endothelial growth factor A, and pyruvate dehydrogenase kinase 1, whose expression is directly regulated through the HIF-1 pathway (*Bernhardt et al., 2009*; *Bujaldon et al., 2019*; *Forsythe et al., 1996*; *Zhang et al., 2018*). We performed a total RNA extraction from 100 mg of liver tissue using TRIzol reagent (Invitrogen, Thermo Fisher Scientific, Carlsbad, CA, USA) according to the manufacturer's specifications. The RNA was quantified using a Microdrop Multiskan GO (Thermo Fisher Scientific, Carlsbad, CA, USA) and stored at −80 °C until analysis.

We performed all the RT-qPCRs using the GoTaq 1-Step kit (Promega Corporation, Madison, WI, USA). The gene coding for β-actin (*Actb*) was used as the housekeeping gene. The following primers were used: *Hmox1* forward 5′-GCCTGCTAGCCTGGTTCAAGA-3′, *Hmox1* reverse 5′-GAGTGTGAGGACCCATCGCA-3′, *Vegfa* forward: 5′-CCGTCCTGTGTGCCCCTAAT-3′, *Vegfa* reverse: 5′-AAACAAATGCTTTCTCCGCT-3′, *Pdk1* forward: 5′-GATTGCCCATATCACGCCTCT-3′, *Pdk1* reverse: 5′-CTCGTGGTTGGT TCTGTAATGC-3′, *Actb* forward 5′-CCCTGGCTCCTAGCACCAT-3′, and *Actb* reverse 5′-GATAGAGCCACCAATCCACACA-3′. We performed every reaction according to the manufacturer's specifications, using 200 ng of RNA for each reaction and the primers at a concentration of 100 nM, to complete a final volume of 20 μL. The following reaction conditions were used: one cycle of reverse transcription at 37 °C for 15 min, one cycle of reverse transcription inactivation and Go Taq DNA Polymerase activation at 95 °C for 10 min, and 40 cycles of denaturation at 95 °C for 10 s and annealing/extension at 60 °C for 30 s. Fold changes of gene expression from the SH group were calculated using the $2^{-\Delta\Delta Ct}$ method.

## Statistical analysis

Data were analyzed using a Shapiro–Wilk normality test, followed by either a one-way ANOVA test with a Tukey *post hoc* test or a Kruskal-Wallis test with a Dunn *post hoc* test. For the gene expression assessment, the $-\Delta\Delta C_T$ values were analyzed. All the statistical analyses were performed in R v. 4.0.1, using the packages *tidyverse*, *cowplot*, and *PMCMRplus*. The full datasets are supplied in Supplemental Informations 1–4, while the code used for the statistical analysis is supplied in Supplemental Information 5. The results are expressed as mean ± standard deviation or median (interquartile range), depending on their distribution. Differences between groups are considered significant at $p < 0.05$.

# RESULTS

## (*S*)-2HG was successfully synthesized from L-glutamic acid

(*S*)-2HG was produced in a 29.78% yield after alkaline hydrolysis, and it was obtained as a beige powder, highly hygroscopic, and soluble in water but insoluble in organic solvents such as methanol, ethanol, acetone, and ethyl acetate. The $^1$H-NMR spectrum showed the

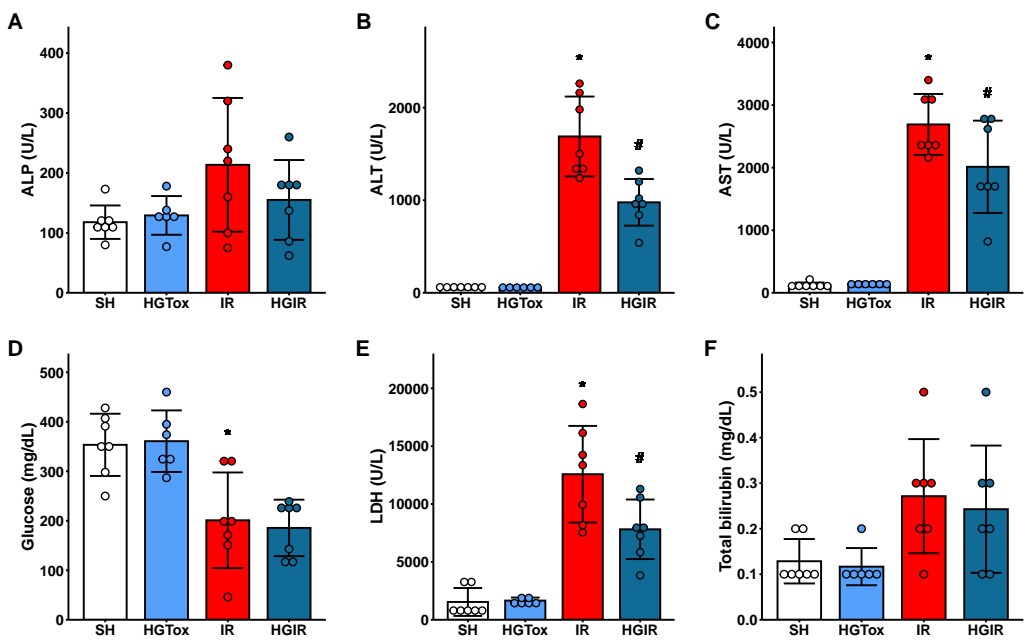

**Figure 1** **Effect of the administration of sodium (S)-2-hydroxyglutarate on the biochemical markers.**
(A) Effect on the serum activity of ALP; (B) effect on the serum activity of ALT, $\star p < 0.0001$ *vs* SH group, #$p = 0.0002$ *vs* IR group; (C) effect on the serum activity of AST, $\star p < 0.0001$ *vs* SH group, #$p = 0.0465$ *vs* IR group; (D) effect on the serum concentration of glucose, $\star p = 0.0032$ *vs* SH group, ‡$p = 0.0012$ *vs* SH group; (E) effect on the serum activity of LDH, $\star p < 0.0001$ *vs* SH group, #$p = 0.0109$ *vs* IR group; (F) effect on the serum concentration of total bilirubin. One-way ANOVA test with Tukey *post hoc* test in (A-E) Kruskal–Wallis test with Dunn *post hoc* test in F. ALP, alkaline phosphatase; ALT, alanine aminotransferase; AST, aspartate aminotransferase; LDH, lactate dehydrogenase. Values expressed as mean ± standard deviation.

following signals: 1.82 ppm (1H, m); 1.96 ppm (1H, m); 2.22 ppm (2H, m), and 4.00 ppm (1H, dd, $J' = 7.6$ Hz, $J'' = 4.0$ Hz) (Fig. S1). In the $^{13}$C-NMR spectrum, the following signals were observed: 33.91, 36.38, 75.01, 184.10, and 185.71 ppm (Fig. S2). Both spectra were consistent with the reported in the literature for this compound (*Bal & Gryff-Keller, 2002*; *Cienfuegos-Pecina et al., 2020*). A value of $[\alpha]_D\ 20° = 8.4°$ cm$^3$g$^{-1}$dm$^{-1}$ was observed, and it was consistent with previously reported data (*Cienfuegos-Pecina et al., 2020*; *Ritthausen, 1872*).

## (S)-2HG is hepatoprotective, but not hepatotoxic at the tested dose

To evaluate whether (S)-2HG produces an acute hepatotoxic effect, we compared the levels of liver injury biomarkers between the SH and HGTox groups. We did not observe a significant difference between these groups in any of the assessed biochemical markers (Fig. 1).

To assess whether (S)-2HG had a hepatoprotective effect, we compared the levels of the liver injury biomarkers among the SH, IR, and HGIR groups. Neither the treatment with (S)-2HG nor the IR injury induction affected the liver weight of the animals (Fig. S3). The induction of IR injury significantly increased the serum activities of ALT, AST, and

**Table 1  Effect of the administration of sodium (S)-2-hydroxyglutarate on the oxidative stress biomarkers.**

| Biomarker | SH | HGTox | IR | HGIR |
|---|---|---|---|---|
| MDA (nmol/mg of protein) | $651.8 \pm 89.8$ | $745.5 \pm 102.8$ | $594.8 \pm 46.6$ | $615.7 \pm 61.8$ |
| SOD (IU/mg of protein) | $23.19 \pm 3.15$ | $27.94 \pm 7.33$ | $26.28 \pm 4.98$ | $24.58 \pm 9.53$ |
| GPx (nmol/min/mg of protein) | $831.0 \pm 91.3$ | $840.2 \pm 95.4$ | $783.3 \pm 32.6$ | $722.9 \pm 44.2^{*}$ |

Notes.

One-way ANOVA test, Tukey post hoc test.

MDA, malondialdehyde; SOD, superoxide dismutase; GPx, glutathione peroxidase.

$^{*}p = 0.0396$ *vs* SH group.

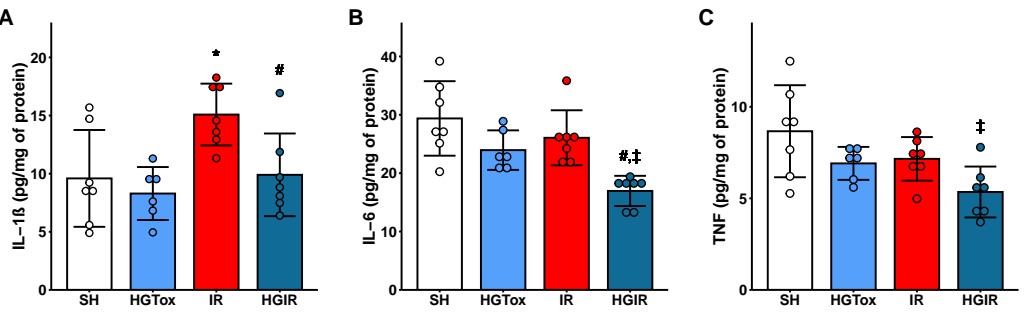

**Figure 2  Effect of the administration of sodium (S)-2-hydroxyglutarate on the proinflammatory cytokines.** (A) Effect on the tissue concentration of IL-1β, $^{*}p = 0.0228$ *vs* SH group, $^{#}p = 0.0336$ *vs* IR group; (B) effect on the tissue concentration of IL-6, $^{#}p = 0.0054$ *vs* IR group, $^{‡}p = 0.0002$ *vs* SH group; (C) effect on the tissue concentration of TNF, $^{‡}p = 0.0050$ *vs* SH group. One-way ANOVA test with Tukey *post hoc* test. IL-1β, interleukin 1β; IL-6, interleukin 6; TNF, tumor necrosis factor. Values expressed as mean ± standard deviation.

LDH compared to the SH group, while the treatment with (S)-2HG produced a significant decrease in the serum activities of these enzymes compared to the IR group (Fig. 1). A significant decrease in serum glucose concentration was observed in both, the IR and HGIR groups compared to the SH group. No significant differences were observed among groups in the serum activity of ALP and the serum concentration of total bilirubin (Fig. 1).

## The experimental model of total liver IR injury did not affect the levels of oxidative stress biomarkers in liver tissue

No significant differences were observed in the levels of oxidative stress biomarkers between the IR and SH groups (Table 1). Their levels were not affected by the administration of (S)-2HG in the HGTox group. We observed a decrease in the tissue activity of GPx in the HGIR group compared with the SH group (Table 1).

## (S)-2HG modulates the concentration of proinflammatory cytokines

Our model of acute liver injury produced a significant increase in the tissue concentration of IL-1β in the IR group compared to the SH group. Treatment with (S)-2HG prevented this increase, as we observed a concentration of IL-1β in the HGIR group significantly lower than the concentration in the IR group (Fig. 2).

The induction of liver IR injury did not affect the tissue concentrations of IL-6 and TNF, however, a significant decrease in the tissue concentration of these cytokines was observed in the HGIR group, compared with both IR and SH groups (Fig. 2). The administration of (*S*)-2HG to the HGTox group did not cause any alteration in the levels of the assessed cytokines compared to the SH group (Fig. 2).

### (*S*)-2HG produced a trend to decrease the severity of the histological liver injury, but it was not significative

The administration of (*S*)-2HG to the HGTox group did not produce a significant change in the scores of histological liver injury compared to the SH group (Table 2, Fig. 3). On the other hand, our model of IR injury induced a significant degree of necrosis and sinusoidal congestion in the IR group compared to the SH group, without producing a significant increase in the degree of cytoplasmic vacuolization (Table 2, Fig. 3). The histopathological evaluation of the IR group showed cellular eosinophilia and nuclear hyperchromasia. Cellular necrosis was severe at zone 3 of the liver acinus, with disruption of the central vein endothelial border (Fig. 3). The administration of (*S*)-2HG tended to decrease the severity of necrosis and sinusoidal congestion compared to the IR group, although this decrease was not statistically significant (Table 2, Fig. 3).

### The treatment with (*S*)-2HG decreased the expression of *Vegfa* and *Pdk1* in liver tissue, without affecting the expression of *Hmox1*

Contrarily as we expected, the treatment with (*S*)-2HG did not increase the expression of *Hmox1, Vegfa,* or *Pdk1* in liver tissue. The expression of *Hmox1* in liver tissue was not affected by the administration of (*S*)-2HG or by the induction of IR injury (Fig. 4). On the other hand, we observed a significant decrease in the expression of both, *Vegfa* and *Pdk1* in the rats of the HGTox group compared to the SH group (Fig. 4). The induction of IR also decreased the expression of these genes compared to the SH group (Fig. 4). These observations suggest that the hepatoprotective effect of (*S*)-2HG is not mediated by the HIF-1 metabolic axis, since we did not observe the genetic footprint of the HIF-1α stabilization, indicating the involvement of a different, currently unidentified mechanism.

## DISCUSSION

Liver transplantation is still the definitive therapy for end-stage liver diseases. However, the lack of donors makes every organ invaluable. Hence, a plethora of strategies has been assessed to improve graft survival. Several strategies have been developed to ameliorate IR injury, aiming to stop the damage cascade at several levels. Pretreatment with antioxidant compounds, such as curcumin or plant extracts with free-radical scavenging activity, has been used to decrease the intensity of the oxidative burst during IR injury (*Jiménez Pérez et al., 2016*; *Shen et al., 2007*; *Torres-González et al., 2018*; *Yildiz et al., 2015*). The use of anti-inflammatory drugs has also been evaluated, aiming to ameliorate the inflammatory response after IR injury, which is the pathogenic mechanism that drives acute graft rejection (*Abdel-Gaber et al., 2015*). These strategies are aimed to ameliorate the damage that happens during reperfusion. However, only a few have been designed to prevent during the ischemia
**Table 2  Histological evaluation after administration of sodium (*S*)-2-hydroxyglutarate.**

| Parameter | SH | HGTox | IR | HGIR |
|---|---|---|---|---|
| | 0 | 1 | 3 | 3 |
| | 0 | 1 | 2 | 3 |
| Cell | 0 | 0 | 3 | 1 |
| necrosis | 0 | 0 | 3 | 3 |
| | 0 | 1 | 3 | 2 |
| | 0 | 2 | 3 | 1 |
| | 0 | ND | 3 | 2 |
| Median (Interquartile range) | 0.00 (0.00–0.00) | 1.00 (0.25–1.00) | 3.00 (3.00–3.00)[*] | 2.00 (1.50–3.00) |
| | 1 | 1 | 3 | 1 |
| | 2 | 2 | 3 | 2 |
| Sinusoidal | 1 | 0 | 2 | 2 |
| congestion | 0 | 1 | 2 | 2 |
| | 0 | 2 | 3 | 2 |
| | 0 | 0 | 3 | 1 |
| | 1 | ND | 3 | 0 |
| Median (Interquartile range) | 1.00 (0.00–1.00) | 1.00 (0.25–1.75) | 3.00 (2.50–3.00)[**] | 2.00 (1.00 -2.00) |
| | 0 | 0 | 0 | 1 |
| | 0 | 0 | 1 | 1 |
| Cytoplasmic | 0 | 0 | 0 | 0 |
| vacuolization | 0 | 1 | 0 | 0 |
| | 0 | 1 | 1 | 2 |
| | 0 | 0 | 3 | 1 |
| | 0 | ND | 0 | 0 |
| Median (Interquartile range) | 0.00 (0.00–0.00) | 0.00 (0.00–0.75) | 0.00 (0.00–1.00) | 1.00 (0.00–1.00) |

**Notes.**
Kruskal–Wallis test, Dunn *post hoc* test.
[*]$p = 0.0002$ *vs* SH group.
[**]$p = 0.0021$ *vs* SH group.
The whole dataset is shown in the table.

the cascade of events that lead to the reperfusion injury. Surgical techniques such as IPC and RIPC have been demonstrated to be promising in pre-clinical studies (*Abu-Amara et al., 2011*), however, the evidence of their effectiveness in clinical studies is controversial (*Zapata-Chavira et al., 2019*; *Zapata-Chavira et al., 2015*). The mechanism behind IPC and RIPC is still not completely understood, but it is well known that the HIF-1 pathway plays an essential role in its protective effect (*Cai et al., 2013*). The stabilization of HIF-1α through the inhibition of the EGLN family of prolyl-4-hydroxylases has demonstrated a protective effect against IR injury in several tissues (*Hill et al., 2008*; *Vogler et al., 2015*). In a previous report of our laboratory, we demonstrated that (*S*)-2HG has a nephroprotective effect against IR injury in Wistar rats (*Cienfuegos-Pecina et al., 2020*).

Several animal models have been used to study the mechanisms of IR injury in several organs. Studies relying on relatively big animals, such as pigs (*Andria et al., 2013*) and

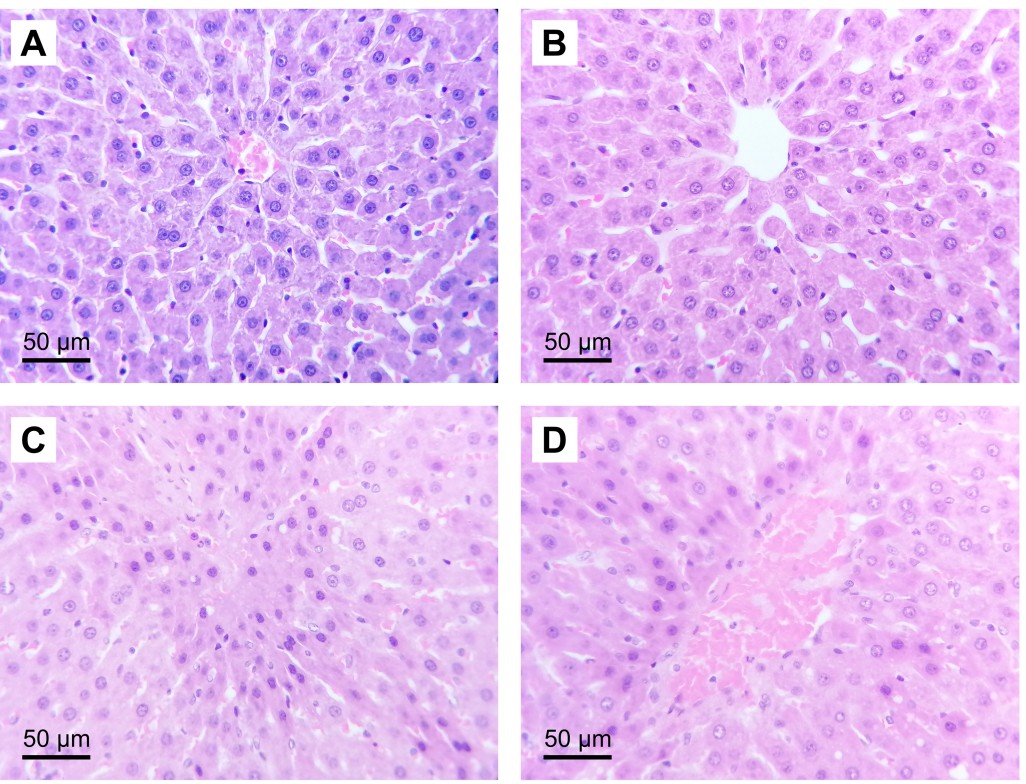

**Figure 3** **Representative liver micrographs of the experimental groups.** Hematoxylin and eosin staining (original magnification: 400×). (A) SH group, (B) HGTox group. Tissue architecture was conserved in A and B, without significant cell necrosis or sinusoidal congestion. (C) IR group, (D) HGIR group. Severe cellular necrosis is observed in zone 3 of the liver acinus in C, while the severity of the damage was decreased in D.

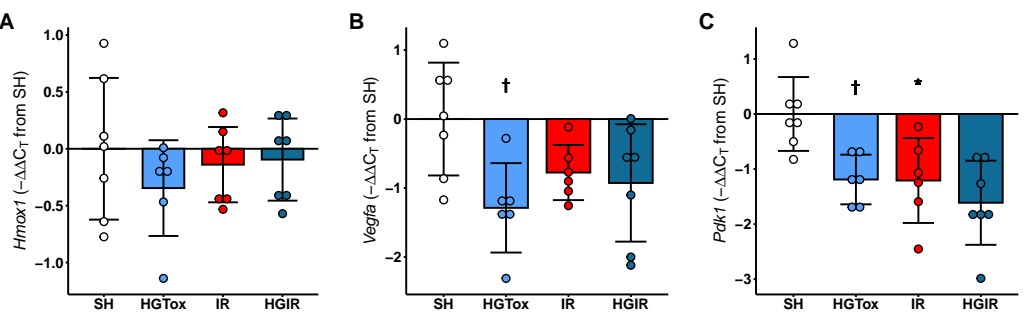

**Figure 4** **Effect of the administration of sodium (*S*)-2-hydroxyglutarate on the expression of genes regulated by the HIF-1 pathway in liver tissue.** (A) Effect on the tissue expression of *Hmox1*; (B) effect on the tissue expression of *Vegfa*, †$p = 0.0185$ *vs* SH group; (C) effect on the tissue expression of *Pdk1*, †$p = 0.0227$ *vs* SH group, *$p = 0.0204$ *vs* SH group. One-way ANOVA test with Tukey *post hoc* test. Values expressed as mean ± standard deviation.

dogs (*Choi et al., 2010*) have the advantage of a good similarity to humans, however, these models have been gradually abandoned due to ethical considerations. Experimental models in rodents, such as mice, are considered attractive, given the availability of genetically modified strains. On the other hand, rat models of IR injury have been well characterized (*Jiménez Pérez et al., 2016*) and have the advantage of larger anatomical structures, allowing the performance of accurate surgical procedures without the need for microsurgery instruments. In addition, rat models are a good starting point for the screening of compounds with potential hepatoprotective activity. Hence, we decided to perform our study using a well-characterized model of liver IR injury in Wistar rats.

It has been demonstrated that the severity of IR injury is sex-dependent, involving a modulatory effect of the sex hormones. Some studies have shown that female rodents exert a higher resistance to liver IR injury than males, *via* an estrogen-dependent mechanism (*Eckhoff et al., 2002*; *Harada et al., 2003*). Estrogen induces the activity of the endothelial cell isoform of NO synthase (eNOS) through the estrogen receptor-α. The mechanism involves either the Akt-mediated activation of eNOS or the upregulation in the transcription of its gene (*Harada et al., 2003*). Since the effect of IR injury is different between females and males, the study of mixed groups is not appropriate. Hence, we choose to assess the effect of our liver IR injury model specifically in female Wistar rats.

In this study, we observed that the pretreatment with (*S*)-2HG at a dose of 25 mg/kg protects the liver against IR in a protocol of 20 min of ischemia and 1 h of reperfusion, evidenced by a significant decrease in the serum activities of ALT, AST, and LDH compared to the IR group. These three enzymes are the most used biomarkers to assess hepatocellular necrosis in a clinical context because are highly expressed in the hepatocytes (*Giannini, Testa & Savarino, 2005*; *Jiménez Pérez et al., 2016*). We also observed a decrease in the serum concentrations of glucose in both, the IR and HGIR groups compared to SH. It has been demonstrated that liver ischemia impairs glucose metabolism, affecting the gluconeogenesis pathway and decreasing the serum glucose concentration during the late ischemia and reperfusion (*Bailey & Reinke, 2000*; *Bloechle et al., 1994*). There was no significant increase, but a trend, in the serum activity of ALP and the serum concentration of total bilirubin after the induction of IR injury, indicating that IR injury induction affected to a lesser extent the bile ducts than the liver parenchyma. On the other hand, (*S*)-2HG did not affect the levels of the assessed biochemical markers compared in the HGTox group compared to the SH group, evidencing that the compound has no hepatotoxic effect at the tested dose. These results agree with our previous report, in which no acute hepatotoxic or nephrotoxic effect was observed at a dose of 12.5 and 25 mg/kg (*Cienfuegos-Pecina et al., 2020*), however, additional studies assessing the toxicity of (*S*)-2HG at higher doses and after chronic exposure are still needed.

Despite the involvement of oxidative stress in the mechanism of IR injury, we did not observe a significant difference in the assessed oxidative stress biomarkers (MDA, SOD, and GPx) neither in the IR group compared to the SH group, nor in the HGIR group compared to the IR group. It has been reported in the literature that despite the role of oxidative stress in the process of IR injury, its effect on the three assessed biomarkers is detected only after longer periods of ischemia and reperfusion (*Gupta et al., 1997*). The

characteristics of our model (20 min of ischemia and 1 h of reperfusion) allow us to induce a moderated IR injury, however, the assessed biomarkers were not sensitive enough to detect the oxidative stress associated with IR injury. Even when this model is ideal for the screening of new compounds with hepatoprotective activity, additional experiments must be performed to accurately quantify the effect of the evaluated compounds on the redox balance after the induction of IR injury.

Proinflammatory cytokines play an important role in the physiopathology of IR injury, being involved in the acute rejection of the graft. It has been demonstrated that the intensity of IR injury dictates the magnitude of the inflammatory response, as evidenced by the correlation of the expression of IL-1β and the duration of ischemia and reperfusion (*Jiménez-Castro et al., 2019*). In this study, we observed that only IL-1β was significantly affected by our model of IR injury. This agrees with a previous report showing that a mild ischemic injury is not enough to induce an increase in the tissue concentration of proinflammatory cytokines during the reperfusion period (*Jiménez Pérez et al., 2016*). We observed a significant decrease in the tissue concentration of IL-1β in the HGIR group compared to the IR group. This indicates that the amelioration of IR injury by the treatment with (*S*)-2HG is enough to impact the inflammatory response derived from the ischemic insult. Besides, despite our model of IR injury was unable to increase the tissue concentrations of IL-6 and TNF in the IR group compared to the SH group, we observed a decrease in the levels of these cytokines in the HGIR group.

The induction of IR injury caused a significant disruption of the tissue architecture, evidenced by severe cell necrosis and sinusoidal congestion in zone 3 of the liver acinus. Since zone 3 is the last to be reached by the oxygenated blood coming from the portal triad, it is more prone to be affected after the induction of IR injury (*Ali et al., 2015*). In our study, our model of mild IR injury produced significant damage in the perivenular zone, without affecting extensively the region around the portal triad. This histopathological finding agrees with the levels of serum ALP and total bilirubin in the IR group, which did not increase significantly compared to the SH group. The administration of (*S*)-2HG to the HGIR group produced a decrease in the severity of both, cell necrosis and sinusoidal congestion. However, this decrease was not statistically significant in the non-parametric Kruskal-Wallis test. A quantitative morphometric analysis of the tissue sections would be helpful in future studies to measure the effect of (*S*)-2HG at the tissue level.

Heme oxygenase 1 is the inducible isoform of heme oxygenase, and it is codified by the gene *Hmox1*. Its expression is directly regulated by the HIF-1 pathway (*Lee et al., 1997*), just like the genes codifying for the vascular endothelial growth factor A (*Vegfa*) (*Forsythe et al., 1996*) and the pyruvate dehydrogenase kinase 1 (*Pdk1*) (*Majmundar, Wong & Simon, 2010*). Together, these genes are assessed to detect the footprint of the stabilization of HIF-1α, since this protein is not easily detected directly, mainly due to its instability in normoxic conditions (*Wang et al., 2017*; *Zhang et al., 2018*). A previous report demonstrates that the upregulation of *Hmox1* by the stabilization of HIF-1α attenuates postischemic myocardial injury in both, *in vitro* and *in vivo* models, with an associated downregulation of the expression of IL-8 (*Dawn & Bolli, 2005*). We previously observed that the administration of (*S*)-2HG at a dose of 25 mg/kg induces a 14.15-fold increase in the expression of *Hmox1*

in kidney tissue, with a nephroprotective effect against IR injury (*Cienfuegos-Pecina et al., 2020*). Surprisingly, in this study, we did not observe any modification in the expression levels of *Hmox1* after the administration of 25 mg/kg of (*S*)-2HG. In addition, instead of observing an increase in the expression of *Vegfa* and *Pdk1* after the treatment with (*S*)-2HG, we observed a significant decrease. Also, the induction of IR injury decreased the expression of both genes.

The behavior of *Vegfa* in the rats subjected to IR injury agrees with a previous report, showing that during the first stage of the IR injury induction the expression of *Vegfa* is decreased (*Bujaldon et al., 2019*), however, the decrease of its expression in the HGTox group suggests that the HIF-1—VEGF axis is not involved in the hepatoprotective effect of (*S*)-2HG since *Vegfa* expression is directly upregulated by HIF-1 (*Forsythe et al., 1996*). This finding is supported by the behavior of *Pdk1*, which also significantly decreased its expression in the HGTox group. Together, these results suggest that (*S*)-2HG exerts its hepatoprotective effect by a different mechanism than the one observed in kidney tissue (*Cienfuegos-Pecina et al., 2020*). Further studies are needed to identify the metabolic pathways involved.

The increased activity of (*S*)-2HG dehydrogenase in liver tissue may play a major role in its pharmacokinetics. In the liver tissue, the (*S*)-2HG dehydrogenase has an activity more than 4 times higher than in kidney tissue (*Jansen & Wanders, 1993*). This enzyme catalyzes the oxidation of (*S*)-2HG in a $FAD^+$-dependent mechanism, producing $\alpha$-ketoglutarate (*Rzem, VanSchaftingen & Veiga-da Cunha, 2006*), which, according to a previous report, exhibits a hepatoprotective effect by regulating the polarization of Kupffer cells from an M1 to an M2 phenotype, decreasing the severity of the IR injury and the expression of IL-6 (*Cheng et al., 2019*). The hypothesis of an involvement of an α-ketoglutarate-dependant mechanism would agree with our findings since we observed a downregulation of the tissue concentration of IL-6 in the HGIR group. Because neither SOD nor GPx exhibit $FAD^+$-dependent mechanisms, an increase in the cellular concentration of $FADH_2$ would not affect these biomarkers. Also, it has been reported that α-ketoglutarate inhibits the expression of *Vegfa* in osteosarcoma cell lines (*Kaławaj et al., 2020*), which agrees with our observation that *Vegfa* was downregulated in the rats treated with (*S*)-2HG. The fact that α-ketoglutarate is the actual substrate of the EGLN hydroxylases could also explain the observed downregulation of *Vegfa* and *Pdk1* by inactivation (instead of stabilization) of HIF-1α. Additional experiments are needed to elucidate the mechanism of action of (*S*)-2HG against IR injury in liver tissue.

The pharmacokinetics of (*S*)-2HG is still not fully understood. Since the disodium salt of (*S*)-2HG is highly polar, it is very unlikely that it could permeate through cell membranes. However, it has been reported that (*S*)-2HG undergoes protonation and then cyclization at an acidic pH (*Bal & Gryff-Keller, 2002*). This process would form an equilibrium between (*S*)-2HG and its more hydrophobic lactone in the stomach after its administration *p. o.*, increasing the absorption and biodisponibility. Further studies with a diversity of derivatives of (*S*)-2HG are needed to give us a better understanding of its pharmacokinetics and to identify analogs with a higher pharmacologic potential.

## CONCLUSIONS

(*S*)-2HG has a hepatoprotective effect against IR injury, which involved the amelioration of liver injury biomarkers and proinflammatory cytokines. The administration of (*S*)-2HG did not induce an acute hepatotoxic effect at the tested dose. The expression of *Hmox1* in liver tissue was not affected by the pre-treatment with (*S*)-2HG, while the expression of *Vegfa* and *Pdk1* was decreased, indicating that the HIF-1 pathway does not play a role in the hepatoprotective effect of (*S*)-2HG and suggesting the involvement of additional pathways in its mechanism of hepatoprotection.

### Funding
The authors received no funding for this work.

### Competing Interests
The authors declare there are no competing interests.

### Author Contributions

- Eduardo Cienfuegos-Pecina conceived and designed the experiments, performed the experiments, analyzed the data, prepared figures and/or tables, authored or reviewed drafts of the paper, and approved the final draft.
- Diana P. Moreno-Peña and Liliana Torres-González conceived and designed the experiments, performed the experiments, authored or reviewed drafts of the paper, and approved the final draft.
- Diana Raquel Rodríguez-Rodríguez conceived and designed the experiments, performed the experiments, analyzed the data, authored or reviewed drafts of the paper, and approved the final draft.
- Diana Garza-Villarreal, Oscar H. Mendoza-Hernández, Raul Alejandro Flores-Cantú and Brenda Alejandra Samaniego Sáenz performed the experiments, authored or reviewed drafts of the paper, and approved the final draft.
- Gabriela Alarcon-Galvan performed the experiments, analyzed the data, prepared figures and/or tables, authored or reviewed drafts of the paper, contributed reagents, materials, analysis tools, and approved the final draft.
- Linda E. Muñoz-Espinosa conceived and designed the experiments, authored or reviewed drafts of the paper, contributed reagents, materials, analysis tools, and approved the final draft.
- Tannya R. Ibarra-Rivera and Paula Cordero-Pérez conceived and designed the experiments, performed the experiments, analyzed the data, authored or reviewed drafts of the paper, contributed reagents, materials, analysis tools, and approved the final draft.
- Alma L. Saucedo conceived and designed the experiments, performed the experiments, analyzed the data, prepared figures and/or tables, authored or reviewed drafts of the paper, contributed reagents, materials, analysis tools, and approved the final draft.

## Animal Ethics

The following information was supplied relating to ethical approvals (i.e., approving body and any reference numbers):

The Ethics and Research Committee of the School of Medicine, Universidad Autónoma de Nuevo León. Register number HI19-00002.

## Data Availability

The raw data are available in the Supplementary Files.

## Supplemental Information

Supplemental information for this article can be found online at http://dx.doi.org/10.7717/peerj.12426#supplemental-information.

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
