# Peer review of "Treatment with sodium (S)-2-hydroxyglutarate prevents liver injury in an ischemia-reperfusion model in female Wistar rats"

_PeerJ, doi:10.7717/peerj.12426_

## Round 0.1 · original submission · Major Revisions

· Academic Editor

Major Revisions

As you will see, both reviewers highlighted similar concerns, including a lack of data showing engagement of the HIF signaling pathway. It is likely that rectifying this will require additional experimentation. Both reviewers also asked for justification of the choice to only study female mice. Minor concerns regarding data presentation and statistics were also raised, and should be addressed if you choose to revise and resubmit.

Reviewer 1 ·

Basic reporting

The manuscript has a good flow and is easy to understand. However, there are places where the sentence construction can be improved. Examples include line #s 112, 161, 165, 168, 322, 401 etc.

The authors refer to S-2-HG as an oncometabolite. It is in fact R-2-HG (D isomer) that is the oncometabolite and S-2-HG (L isomer) is actually a hypoxic metabolite. It is requested that authors rectify the discussion surrounding this in the introduction section

Experimental design

The authors describe the experimental protocols and procedure is great detail and provide the raw data sets for their results, which is great to see. I have the below questions/concers about the experimental design

1) Why did the authors use only female rats and not males? The authors do not provide any justification for this choice. Liver injury or liver transplant is not a sex specific pathology and I am not aware of any statistics that suggest that the incidence/risk factor is higher in females. As such, both sexes should be evaluated in the study.

2) The authors use S-2-HG as a inhibitor of EGLN family prolyl hydroxylase to sustain levels of hypoxia-inducible factor (HIF). However, the authors do not evaluate prolyl hydroxylation levels in presence and absence of S-2-HG to ensure that there is target engagement by inhibitor. Similarly, there has been no evaluation of the HIF levels in control, IR and treatment+IR groups. This is a major shortcoming of the study.

3) The authors cite the previous use of S-2-HG for its beneficial effects in nephroprotection from IR injury. This study is from the same group as the authors of the current study. However, a sodium salt of S-2-HG is very hydrophilic and is unlikely to cross the lipophilic cell membranes to exert its inhibitory effect on prolyl hydroxylases. It is therefore possible that S-2-HG is exerting the limited hepatoprotective effects through some other mechanism, which the authors need to investigate.
To claim that S-2-Hg/prolyl hydroxylase/HIF axis has been enganged, the authors need to provide enzyme activity and/or western blot data showing S-2-HG inhibits the intended target.
Also it is suggested that authors synthesize and use a dimethyl ester form of S-2-HG which is likely to much much more membrane permeable than the disodium salt.

4) Authors use only liver injury and inflammation biomarkers to assess the effect of S-2-HG in their model of IR injury. No infarction or functional recovery data after S-2-HG treatment has been provided to show beneficial effect. Authors are requested to generate this data.

Validity of the findings

The statistical analysis on some data needs to be reevaluated. For example in Figure 1c, the difference in AST levels between IR and HGIR groups is unlikely to be significant.

The figure legends need to provide information regarding the reported error bars i.e. they represent standard deviation and not standard error.

Reviewer 2 ·

Basic reporting

Cienfuegos-Pecina et al present an original research article titled “Treatment with sodium (S)-2-hydroxyglutarate prevents liver injury in an ischemia-reperfusion model in wistar rats.” Previous studies have shown that preconditioning and the potential activation of hypoxia-inducible factors (HIF) protect against ischemia-reperfusion injury. The authors proposed that inhibition of EGLN by sodium (S)-2-hydroxyglutarate, resulting in HIF activation attenuates ischemia-reperfusion injury. Hepatic ischemia-reperfusion injury occurs due to anoxic liver injury after major resections and transplantations or in response to systemic hypoxia. The manuscript is clearly written with detailed statistical analysis, sample size calculations and easy to follow graphical presentations.

Experimental design

1. In the introduction the authors state that the rationale for using (S)-2HG is the inhibitory effect on EGLN, which potentially activates HIF and attenuates ischemia-reperfusion injury. The authors do not provide any data on EGLN1-3 activity or HIF1/2. Without these additional measurements it is not possible to assess the validity of the hypothesis. (S)-2HG can be converted to alpha-ketoglutarate and potentially used as a carbon fuel in the Krebs cycle independent of HIF regulation. The authors need to expand their experimental model to account for these factors.

2. Reperfusion injury is a dynamic process that involves immune cell activation and cytokine release, as well as reactive oxygen species. The authors focus on clinically relevant liver function tests that can be assessed in the blood (e.g., ALT, AST, cytokine release). However, other processes that have been implicated in hepatic ischemia-reperfusion injury are not included. For example, anaerobic metabolism, mitochondrial function, intracellular calcium overload and kupfer cell activation are not investigated. (S)-2HG has been shown to affect the activities of mitochondrial dehydrogenases and overall redox state of cells (see Intlekofer AM 2017;13:494-500). The authors need provide additional data on the metabolic effect of (S)-2HG.

3. The authors need to provide additional data that show the degree of liver injury in the experimental model. For example, histology and measurements of tissue edema and distribution.

4. The authors only use female rats in their study. Please provide a justification for using only one sex.

5. The bar graphs should be replaced with dot blots to show individual data points.

Validity of the findings

The authors do not provide sufficient data to support their initial hypothesis that EGLN inhibition by (S)-2HG protects against hepatic ischemia-reperfusion injury.

Observations made by the authors may be valid and reproducible, but the provided experimental data is not sufficient to support their initial hypothesis.

---

## Round 0.2 · Minor Revisions

· Academic Editor

Minor Revisions

The authors have performed new experiments and added new data, which improves the manuscript greatly. They have also backed-off from the claims regarding HIF activation. Some of the responses (in particular, regarding the study only using female rats) are not great, but are satisfactory given the ongoing pandemic situation.

I have only a couple of minor suggestions for further improvement:

(1) In Table 2, the data are presented as medians with interquartile ranges, and this is a bit confusing. For example, for cytoplasmic vacuolization, in the IR group the IQ range was 0-1 and the median was 0. But, in the HGIR group the IQ range was the same but the median was 1. It would be better to show the individual values in a table, or to list the mean, not the median, in order to see if there are any differences between these groups. Medians are not very suitable, for limited numerical scales (say 1 to 3).

(2) In Figure 4, the choice to present the fold changes on a log2 scale is somewhat misleading in terms of the magnitude of the effect, and the noise inherent in the data. It is quite interesting that some of the genes are down by more than 4-fold in some samples of the HG treatment group. Since these data (delta delta Ct values) are already normalized, it is odd to further normalize them again to the SH group. It would be better to just present the original delta delta Ct values on the graphs.

---

## Round 0.3 · accepted · Accept

· Academic Editor

Accept

All issues addressed. No further comments. Acceptable for publication.